# Validity of Self-Reported Body Mass, Height, and Body Mass Index in Female Students: The Role of Physical Activity Level, Menstrual Cycle Phase, and Time of Day

**DOI:** 10.3390/ijerph16071192

**Published:** 2019-04-03

**Authors:** Eleni Kintziou, Pantelis T. Nikolaidis, Vasiliki Kefala, Thomas Rosemann, Beat Knechtle

**Affiliations:** 1Faculty of Biomedical Sciences, University of West Attica, 12243 Egaleo, Greece; ekintziou@teiath.gr (E.K.); pademil@hotmail.com (P.T.N.); valiakef@teiath.gr (V.K.); 2Exercise Physiology Laboratory, 18450 Nikaia, Greece; 3Gesundheitszentrum St. Gallen, 9001 St. Gallen, Switzerland; Thomas.Rosemann@usz.ch; 4Institute of Primary Care, University of Zurich, 8006 Zurich, Switzerland

**Keywords:** anthropometry, MET, questionnaire, survey, vigorous exercise

## Abstract

A large part of research using questionnaires for female university students relies on self-reported body mass, height, and body mass index (BMI) data; however, the validity of these data in this population group is unknown. Therefore, the aim of the present study is to examine the validity of self-reported body mass, height, and BMI in female students. Female students of biomedical sciences (*n* = 93, age 21.8 ± 4.7 years, height 1.63 ± 0.06 m, weight 60.5 ± 11.9 kg, and BMI 22.7 ± 3.8 kg/m^2^) completed the short version of the International Physical Activity Questionnaire and were tested for anthropometric characteristics at three different times of the day (12–2 p.m., *n* = 36; 2–4 p.m., *n* = 20; 4–6 p.m., *n* = 37). Participants over-reported height (+0.01 ± 0.02 m, +0.9 ± 1.2%, Cohen’s d = 0.22) and under-reported weight (−0.8 ± 2.1 kg, −1.2 ± 3.6%, d = −0.07) and BMI (−0.7 ± 1.0 kg/m^2^, −2.9 ± 4.2%, d = −0.19) (*p* < 0.001). A moderate main effect of time of day on %Δweight (*p* = 0.017, η^2^ = 0.086) and %ΔBMI (*p* = 0.045, η^2^ = 0.067), but not on %Δheight (*p* = 0.952, η^2^ = 0.001), was observed, where the group tested at 4–6 p.m. under-reported weight and BMI more than the 2–4 p.m. group. The weekly metabolic equivalent (MET) × min did not correlate with %Δheight (*r* = 0.06, *p* = 0.657), but its correlations with %Δweight (*r* = −0.27, *p* = 0.051) and %ΔBMI (*r* = −0.238, *p* = 0.089) reached statistical significance. Participants in the early follicular phase reported BMI more accurately (*p* = 0.084, d = 0.68) than those in the mid-luteal phase. In conclusion, female students over-reported height and under-reported weight and BMI. Under-reporting weight and BMI is influenced by time of day and menstrual cycle phase. These findings should be considered by health professionals and researchers when administering questionnaires to female students.

## 1. Introduction

A large part of research in health sciences has relied on the collection of epidemiological data using surveys, including items such as body weight, height, and body mass index (BMI). Weight has been associated with health status, physical activity, body image, and self-esteem [1]. Accordingly, the need to validate these items has been already identified, and several studies have been conducted on this topic, mostly on men [2]. Sex and age have been shown to relate to the validity of these items; for instance, women and older respondents reported their anthropometric characteristics more accurately than men and younger respondents [3]. Consequently, the accuracy of these self-reported measures should be examined specifically for each sex and age group. A recent review concluded that there was an underestimation of weight by −0.9 kg and an overestimation of height by 0.4 cm in women aged from 12 to 49 years [2]. Nevertheless, no information exists on female university students.

University students comprise of a population group with specific characteristics in terms of having large physiological and psychological changes at a time between adolescence and adulthood. During this period of life, university students undergo changes with regard to several aspects such as lifestyle and consumer habits [4,5], which, in turn, may impact daily total energy expenditure and intake, resulting in changes of weight and BMI. In addition, the accuracy of self-reporting anthropometric characteristics may depend on the time of day. Although it has been demonstrated that most physiological characteristics follow a circadian fluctuation [6] and that height decreases during the day [7], no information exists for the circadian variation of weight so far. Nevertheless, it would be reasonable to assume that weight and BMI increases during day due to the drink and food consumption [8]. Moreover, it has been shown that weight fluctuates during the menstrual cycle with minimal and maximal values in the follicular and luteal phase, respectively [9].

Despite the widely used self-reported measures of these anthropometric characteristics in clinical practice and epidemiological studies on female students, limited information exists with regard to the validity of these measures in this population group [10,11]. Knowledge about the accuracy of self-reported weight, height, and BMI in surveys would be of great practical importance for clinical practice and researchers. Furthermore, information about the role of physical activity, time of day, and menstrual cycle on the accuracy of self-reporting these anthropometric characteristics would be of practical value when administering and interpreting a survey. Therefore, the main aim of the present study is to examine the validity of self-reported height, weight, and BMI. A secondary aim is to analyze the role of physical activity, time of day, and menstrual cycle phase on this validity. For the main aim, based on the existing literature, it is hypothesized that female students would over-report height and under-report weight and BMI. Relating to the role of physical activity, the research hypothesis was that more physically active female students would have a better perception of their body and consequently would report their anthropometric characteristics more accurately. Relating to the role of time of day, it was hypothesized that female students tested later would under-estimate their weight and BMI more than those tested earlier. Furthermore, since weight has its minimal values during the follicular phase of menstrual cycle, it was hypothesized that women in this phase would under-estimate their weight and BMI less than those in the luteal phase.

## 2. Materials and Methods

### 2.1. Participants

Participants were female students of the Faculty of Biomedical Sciences, University of West Attica, Egaleo, Greece (*n* = 93, age 21.8 ± 4.7 years, height 1.63 ± 0.06 m, weight 60.5 ± 11.9 kg, and BMI 22.7 ± 3.8 kg/m^2^). Four participants (4.3% of the sample) were under-weight, 67 were normal-weight (72.0%), 18 were overweight (19.4%), and four were obese (4.3%). The study design was in accordance with the Declaration of Helsinki, adopted in 1964 and revised in 2013, and was approved by the local Institutional Review Board (EPL 2018/4).

### 2.2. Procedures 

All participants were informed about the details of the study including potential benefits and risks and provided informed consent. Thereafter, participants completed a short survey including items about basic demographic data (date of birth, day of menstrual cycle, height, and weight), completed the short version of the International Physical Activity Questionnaire (IPAQ) [12], and were tested for anthropometric characteristics. These procedures occurred at three different times of the day (12–2 p.m., *n* = 36; 2–4 p.m., *n* = 20; 4–6 p.m., *n* = 37), corresponding to different university classes. The outcome of the IPAQ that was analyzed was the weekly product of the metabolic equivalent (MET) and minutes, i.e., MET × min. Height and weight were measured with subjects in minimal clothing and barefoot. A bio-electrical impedance analysis weighing scale (BC-558; Tanita, Arlington Heights, IL, USA) was employed for measurement of weight (to the nearest 0.1 kg) and a portable stadiometer (SECA Leicester, UK) for height (0.01 m). Body mass index was calculated as the quotient of weight (kg) to height squared (m^2^). Since the concentration of hormones during menstrual cycle differ maximally between two particular points—early follicular phase, days 5–7, and mid-luteal phase, days 21–22 [13,14,15]—participants in the early follicular phase (*n* = 12) were compared with those in the mid-luteal phase (*n* = 11) for under-reporting weight and height. This classification was preferred to classifications using wider ranges of menstrual days (e.g., follicular phase, days 5–13; luteal phase, days 15–25) [16]. Although the latter classification would lead to increasing the sample size, increasing the range of days would decrease the differences between the concentration of the hormones and, consequently, their potential effect on weight. All participants were also grouped into quartiles (Q1, Q2, Q3, and Q4) according to percentage difference in reporting BMI (%ΔBMI) with Q1 under-reporting BMI the most. %ΔBMI was calculated using the formula 100 × (reported BMI − measured BMI)/measured BMI, with negative or positive scores denoting under-reporting or over-reporting BMI, respectively. A comparison of BMI groups (under-, normal-, over-weight, and obese) was not performed due to large differences in the sample size of each group.

### 2.3. Statistical and Data Analysis

Statistical analyses were conducted using GraphPad Prism v.7.0 (GraphPad Software, San Diego, CA, USA) and IBM SPSS v.23.0 (SPSS, Chicago, IL, USA). Statistical significance was set at *p* = 0.10 due to the relatively small size of sub-groups. Data—for the whole sample and not for each sub-group—were tested for normality using the Kolmogorov–Smirnov test and visual inspection of quantile–quantile plots. Data were presented as mean and standard deviations. A dependent Student’s t-test compared self-reported and measured height, weight, and BMI in the whole sample as well as participants in the early follicular phase with those in the mid-luteal phase. The magnitude of the differences between self-reported and measured scores was examined using Cohen’s d. Bland–Altman analysis was used to examine the accuracy and variability of prediction equations [17]. One-way analysis of variance (ANOVA) examined differences in self-reporting anthropometric characteristics among the three times of day (12–2 p.m., 2–4 p.m., and 4–6 p.m.) and eta squared (η^2^) was used to estimate the magnitude of these differences [18]. Pearson correlation coefficient r examined the relationship between self-reported weight, height, and BMI, and IPAQ in MET × min [19]. %ΔBMI quartiles were also compared for all variables using a one-way ANOVA. A chi-square test (χ^2^) examined the quartile × time of day and quartile × menstrual cycle phase associations, and Cramer’s phi (φ) estimated the magnitude of these associations.

## 3. Results

Participants over-reported height (+0.01 ± 0.02 m, +0.9 ± 1.2%, Cohen’s d = 0.22) and under-reported weight (−0.8 ± 2.1 kg, −1.2 ± 3.6%, d = −0.07) and BMI (−0.7 ± 1.0 kg/m^2^, −2.9 ± 4.2%, d = −0.19) (*p* < 0.001) (Figure 1). A moderate main effect of time of day on %Δweight (*p* = 0.017, η^2^ = 0.086) and %ΔBMI (*p* = 0.045, η^2^ = 0.067), but not on %Δheight (*p* = 0.952, η^2^ = 0.001), was observed, where the group tested at 4–6 p.m. under-reported weight and BMI more than the 2–4 p.m. group (Figure 2). The weekly MET × min correlated with %Δweight (*r* = −0.27, *p* = 0.051) and %ΔBMI (*r* = −0.238, *p* = 0.089), but not with %Δheight (*r* = 0.06, *p* = 0.657). Participants in the early follicular phase reported BMI more accurately (*p* = 0.084, d = 0.68) than those in the mid-luteal phase, but not weight (*p* = 0.104, d = 0.45) and height (*p* = 0.548, d = −0.26).

A medium effect of %ΔBMI on MET × min was observed (*p* = 0.086, η^2^ = 0.127) with Q3 being the most physically active and Q4 the least (Table 1). A quartile × time of day association is shown (χ^2^ = 11.38, *p* = 0.077, φ = 0.35), where at 12–2 p.m. most participants were classified in Q3, at 2–4 p.m. most were in Q4, and at 4–6 p.m. most were in Q1 and Q2. A quartile × menstrual cycle phase association was also found (χ^2^ = 8.31, *p* = 0.040, φ = 0.60), where most of Q1 and Q2 were at the mid-luteal phase, and most of Q3 and Q4 at the early follicular phase.

## 4. Discussion

The main findings of the present study were that (i) height was over-reported, whereas weight and BMI were under-reported; (ii) the 4–6 p.m. group under-reported weight and BMI more than the 2–4 p.m. group; (iii) the weekly MET × min correlated with %Δweight and %ΔBMI; and (iv) those in the early follicular phase were more likely to report an accurate weight and BMI than those in the mid-luteal phase.

The overestimation of height (by ~1%) and underestimation of weight (by ~1%) and BMI (by ~3%) was in agreement with previous findings in university students [10,11]. For instance, American female students over-reported height by 0.5 cm and under-reported weight and BMI by 1.3 kg and 0.6 kg/m^2^, respectively [11]. A potential explanation of the underestimation of weight may be recent weight gain since entering university, which has been identified as a major cause of weight gain in women [20]. Surprisingly, the group who under-reported BMI the most (Q1) did not have the largest BMI, which was in disagreement with previous studies which showed that heavier female students tended to under-report their weight [11,21,22]. This discrepancy may be explained by the absence of many high BMI scores in the participants of the present study. On the other hand, the largest underestimation of BMI was observed previously in individuals at the high end of BMI, particularly at values larger than 28 kg/m^2^ [22].

The findings relating to the role of time of day, physical activity, and menstrual cycle phase confirmed the corresponding research hypotheses. An inverse correlation of small magnitude was observed between weekly MET × min and under-reporting weight and BMI, indicating that physically active students more accurately perceive their body compared to their less active peers. This was in line with previous findings showing that exercise participation resulted in positive body image [23,24]. It should be highlighted that physical activity (weekly MET × min) in the present study was evaluated using the short version of IPAQ, which has been previously shown as a valid and reliable measure [25,26]. Nevertheless, Lee and colleagues [27] identified limitations of interpreting the results obtained via the use of self-reported questionnaires, e.g., a low correlation between data collected with questionnaires and the amount of physical activity performed by participants, and overestimation of the actual physical activity. These limitations should be considered when interpreting the results about physical activity. For future studies, it would be advisable to use objective methods for measuring physical activity, e.g., accelerometers.

In addition, the validity of the self-reported measures was related to the menstrual cycle phase. The discrepancy with previous research showing no relationship relating to menstrual cycle in students 18–24 years [28] should be attributed to the differing choice of specific phases for physiological characteristics [29] to the present study. From a physiological point of view, energy intake and expenditure increases in the luteal phase compared to the follicular phase leading to increased frequency of cravings for foods [30].

A limitation of the present study was that the sample consisted only of students of biomedical sciences. It was assumed that students in allied health professions would be more aware about their anthropometric characteristics than students of other professions. Thus, caution should be taken in generalizing the findings to students of other professions. Moreover, an unspecified aspect that was not addressed in this study which may affect the results was satisfaction or dissatisfaction with one’s body, since body image could condition the perceptions of the measurements of the human body [31]. On the other hand, the strength of the study was its novelty, as, to the best of our knowledge, it was the first to examine this topic with regard to physical activity, time of day, and menstrual cycle. Considering the wide use of self-reported anthropometric characteristics for clinical practice and scientific research, these results would be of great practical and theoretical value, respectively.

## 5. Conclusions

In summary, female students over-reported height and under-reported weight and BMI. Under-reporting weight and BMI is influenced by time of day, and likely by physical activity level (the higher the physical activity, the lower the under-reporting) and menstrual cycle phase (larger under-reporting in the mid-luteal phase). These findings should be considered by health professionals when administering questionnaires to female students. In particular, surveys in female students including items on self-reported height and weight should also have items about physical activity and menstrual cycle phase to improve their validity. Considering the results of the present study, the further use of self-reported height and weight is recommended.

## Figures and Tables

**Figure 1 ijerph-16-01192-f001:**
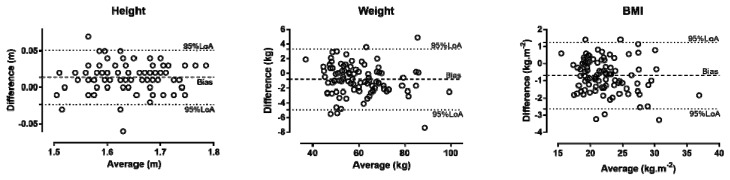
Bland–Altman plots showing the difference (bias) between reported and measured height, weight, and body mass index. BMI = body mass index; LoA = limit of agreement; Difference = reported − measured; Average = (reported + measured)/2.

**Figure 2 ijerph-16-01192-f002:**
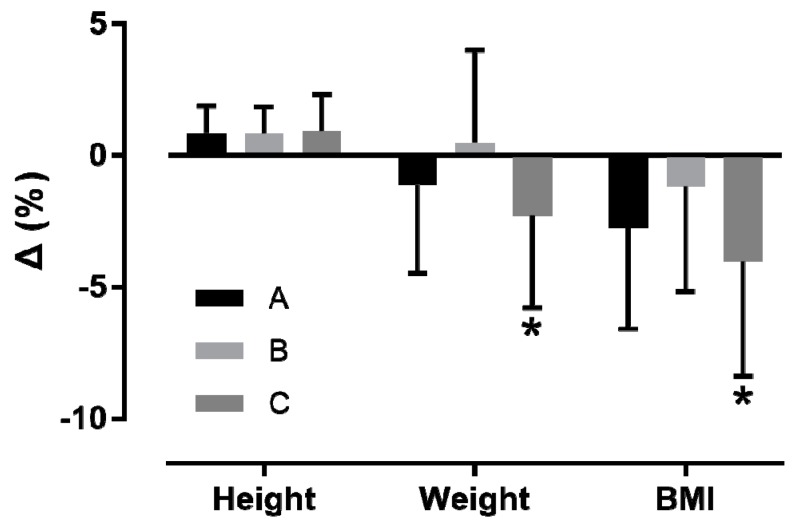
Percentage difference (Δ) between reported and measured height, weight, and body mass index at 12–2 p.m. (group A, *n* = 36), 2–4 p.m. (group B, *n* = 20), and 4–6 p.m. (group C, *n* = 37). BMI = body mass index; * different from group B at *p* < 0.001.

**Table 1 ijerph-16-01192-t001:** Scores of quartile groups based on %ΔBMI.

	Total (*n* = 93)	Q1 (*n* = 23)	Q2 (*n* = 23)	Q3 (*n* = 23)	Q4 (*n* = 24)
Time of day	A:36, B:20, C:37	A:8, B:3, C:12	A:9, B:2, C:12	A:10, B:5, C:8	A:9, B:10, C:5
Age (years)	21.8 ± 4.7	21.3 ± 3.2	21.8 ± 5.0	20.7 ± 1.5	23.3 ± 6.9
IPAQ (MET × min)	2702 ± 2264	3189 ± 2206	2448 ± 2808	3729 ± 2301	1592 ± 1253
Menstrual cycle phase	early FP:12, mid LP:11	early FP:3, mid LP:3	early FP:0, mid LP:5	early FP:4, mid LP:2	early FP:5, mid LP:1
Height					
Measured height (m)	1.63 ± 0.06	1.62 ± 0.06	1.65 ± 0.06	1.62 ± 0.06	1.63 ± 0.07
Reported height (m)	1.64 ± 0.06	1.64 ± 0.06	1.66 ± 0.05	1.63 ± 0.06	1.63 ± 0.08
Δheight (m)	0.01 ± 0.02	0.03 ± 0.02	0.02 ± 0.01	0.01 ± 0.01	0 ± 0.02
%Δheight	0.9 ± 1.2	1.7 ± 1.2	1.0 ± 0.6	0.7 ± 0.8	0.1 ± 1.3
Weight					
Measured weight (kg)	60.5 ± 11.9	61.1 ± 12.5	65.6 ± 13.1	58.2 ± 10.0	57.3 ± 10.8
Reported weight (kg)	59.7 ± 11.6	58.0 ± 12.3	64.0 ± 12.8	58.0 ± 9.7	58.8 ± 11.1
Δweight (kg)	−0.8 ± 2.1	−3.0 ± 1.7	−1.6 ± 0.8	−0.2 ± 1.2	1.5 ± 1.3
%Δweight	−1.2 ± 3.6	−5.1 ± 2.8	−2.4 ± 1.2	−0.2 ± 2.1	2.6 ± 2.2
BMI					
Measured BMI (kg/m^2^)	22.7 ± 3.8	23.3 ± 3.9	24.2 ± 4.4	22.0 ± 3.2	21.5 ± 3.3
Reported BMI (kg/m^2^)	22.0 ± 3.6	21.3 ± 3.6	23.1 ± 4.2	21.7 ± 3.1	22.1 ± 3.4
ΔBMI (kg/m^2^)	−0.7 ± 1.0	−1.9 ± 0.6	−1.1 ± 0.3	−0.4 ± 0.2	0.5 ± 0.4
%ΔBMI	−2.9 ± 4.2	−8.2 ± 2.1	−4.4 ± 0.7	−1.6 ± 0.9	2.4 ± 1.8

Q1, Q2, Q3, and Q4 were quartile groups based on %ΔBMI; %ΔBMI was calculated using the formula 100 × (reported BMI − measured BMI)/measured BMI with Q1 being the quartile under-reporting BMI the most; A, B, and C refer to the time of day group, i.e., 12–2 p.m., 2–4 p.m., and 4–6 p.m., respectively; FP = follicular phase, LP = luteal phase, IPAQ = International Physical Activity Questionnaire, MET = metabolic equivalent.

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
