# Peer review of "Validity of Self-Reported Body Mass, Height, and Body Mass Index in Female Students: The Role of Physical Activity Level, Menstrual Cycle Phase, and Time of Day"

_ijerph, 2019, doi:10.3390/ijerph16071192_

Round 1
Reviewer 1 Report
Overall comments: The introduction is well-organized and clearly establishes the need for the study. The aims are clear. The study was designed appropriately, and the appropriate statistical tests were used. The figures were clear and agreed with the manuscript text.
Line 15: Please clarify that this applies to women, as the wording suggests that you will be examining university students in general.
Line 23: Here in the abstract you mention a 2-4 pm group and a 4-6 pm group. However, these groups have not been mentioned earlier in the abstract, so the reader does not know what these groups are and how they differ. Was this a class time during which the students completed the survey?
Lines 25, 90, 116: Delete "being"; it is colloquial.
Line 26: Delete "likely"; this suggests that you are not sure about the data.
Line 29: Again, why do you use the word "likely"? The data either show that this is true (statistically significant) or not.
Line 30: Delete "been". The large number of errors thus far suggests that this manuscript needs to be carefully proofread by a strong English writer.
Lines 43, 74: There should be a space between a number and a subsequent unit.
Line 45: Change "consist" to comprise.
Line 52: Subject-verb error. Again, I strongly recommend a careful proofreading.
Lines 87-88: Grammar needs corrected.
Lines 142-144: The following sentence is unclear. Please clarify. "However, an unexpected result was that the quartile that 143 under-reported BMI the most (Q1) was not the one with the highest BMI suggesting that BMI per se 144 could not identify the less accurate in their answers participants."
Line 148: Should be "An inverse correlation..."
Line 167: should read "women students"
Line 166: As this study examined the validity of self-reported height and weight on surveys in university students, please add to the conclusion a statement regarding the usefulness of the survey and a more general statement regarding whether or not it is a valid measure.
Author Response
Comments and Suggestions for Authors
Overall comments: The introduction is well-organized and clearly establishes the need for the study. The aims are clear. The study was designed appropriately, and the appropriate statistical tests were used. The figures were clear and agreed with the manuscript text.
Line 15: Please clarify that this applies to women, as the wording suggests that you will be examining university students in general.
Answer: We agree with the expert reviewer and clarified it as suggested.
Line 23: Here in the abstract you mention a 2-4 pm group and a 4-6 pm group. However, these groups have not been mentioned earlier in the abstract, so the reader does not know what these groups are and how they differ. Was this a class time during which the students completed the survey?
Answer: We agree with the expert reviewer and added this information in the abstract („during three times of the day (12-2 pm, n=36; 2-4 pm, n=20; 4-6 pm, n=37)“) and in the methods section.
Lines 25, 90, 116: Delete "being"; it is colloquial.
Answer: We agree with the expert reviewer and deleted it as suggested.
Line 26: Delete "likely"; this suggests that you are not sure about the data.
Answer: We agree with the expert reviewer and deleted it as suggested.
Line 29: Again, why do you use the word "likely"? The data either show that this is true (statistically significant) or not.
Answer: We agree with the expert reviewer and deleted it as suggested.
Line 30: Delete "been". The large number of errors thus far suggests that this manuscript needs to be carefully proofread by a strong English writer.
Answer: We agree with the expert reviewer and deleted it as suggested. In addition, a native speaker revised all text for language.
Lines 43, 74: There should be a space between a number and a subsequent unit.
Answer: We agree with the expert reviewer and revised it as suggested.
Line 45: Change "consist" to comprise.
Answer: We agree with the expert reviewer and corrected it.
Line 52: Subject-verb error. Again, I strongly recommend a careful proofreading.
Answer: We agree with the expert reviewer and corrected it.
Lines 87-88: Grammar needs corrected.
Answer: We agree with the expert reviewer and corrected it („Since hormones’ concentration during menstrual cycle differs maximally between two particular points - early follicular phase, days 5-7, and mid luteal phase, days 21-22 [13-15] - participants being in the early follicular phase (n=12) were compared for underreporting weight and height with those in the mid luteal phase (n=11).“).
Lines 142-144: The following sentence is unclear. Please clarify. "However, an unexpected result was that the quartile that 143 under-reported BMI the most (Q1) was not the one with the highest BMI suggesting that BMI per se 144 could not identify the less accurate in their answers participants."
Answer: We agree with the expert reviewer and deleted it.
Line 148: Should be "An inverse correlation..."
Answer: We agree with the expert reviewer and corrected it as suggested.
Line 167: should read "women students"
Answer: We agree with the expert reviewer and corrected it as suggested.
Line 166: As this study examined the validity of self-reported height and weight on surveys in university students, please add to the conclusion a statement regarding the usefulness of the survey and a more general statement regarding whether or not it is a valid measure.
Answer: We agree with the expert reviewer and added the suggested information („Particularly, surveys in women students including items on self-reported height and weight should also have items about physical activity and menstrual cycle phase to improve their validity. Considering the results of the present study, the further use of self-reported height and weight was recommended.“).
Reviewer 2 Report
The novelty of this work lies in examining the self-reported perception of body mass and height in women university students (age range between 18 and 30 years), a population with a very specific profile that has not been studied in detail until now. The results presented in this work confirms the hypotheses initially proposed, since women university students tend to over-report height and under-report body mass. Comparison between self-perception and real measures was performed taking into account different conditions, such as the times of the day, the physical activity or the menstrual cycle phase.
This study recommends taking into account the deviations of self-evaluation from real measures by health professionals and researchers when administering questionnaires to women university students. The conclusions and their practical application may be interesting for the readerships. However, the use of questionnaires to evaluate several characteristics related to physical activity and its relationship with other physiological parameters presents some controversy in the scientific field.
The results presented in this study are interpreted appropriately and support the research conclusions. Data analysis and interpretation is adequate to the hypotheses proposed since results obtained are able to answer the different questions proposed at the beginning of the work.
However, the text should be improved by including some details explained below.
- Line 80: The document says: “completed the short version of the International Physical Activity Questionnaire (IPAQ) [12] and were tested for anthropometric characteristics. The outcome of IPAQ that was analyzed was the weekly product of metabolic equivalent (MET) and minutes, i.e. MET×min".
In the work carried out by Craig et al (2003), the validation of the IPAQ questionnaire for the quantification of the physical exercise intensity performed by an individual with the METs product and the minutes performed was proposed. In the same way, Strath et al (2013) presents the same methodology to quantify the intensity of exercise carried out based on a questionnaire. However, Lee et al (2011) pointed out the weaknesses of interpreting the results obtained via the use of self-reported questionnaires. Their conclusions stated the low correlation between data collected with questionnaires and the amount of physical activity performed by participants. They showed that self-reported questionnaires overestimate the actual physical activity carried out by participants.
This observation does not directly affect the work done. However, it should be expressed in the main text. It should be clearly explained that the use of questionnaires is a tool for comparison between the different groups defined in each quartile, and it is not a method for quantitatively measure the physical activity carried out by a participant because of the low correlation observed between the self-reported and real measures. For future studies, it would be advisable to use objective methods for measuring physical activity, such as the use of accelerometry.
- Craig CL, Marshall AL, Sjöström M, Bauman AE, Booth ML, Ainsworth BE, et al. International physical activity questionnaire: 12-country reliability and validity. Med Sci Sports Exerc 2003;35:1381–95. doi:10.1249/01.MSS.0000078924.61453.FB.
- Strath SJ, Kaminsky LA, Ainsworth BE, Ekelund U, Freedson PS, Gary RA, et al. Guide to the Assessment of Physical Activity: Clinical and Research Applications A Scientific Statement From the American Heart Association. Circulation 2013:01.cir.0000435708.67487.da. doi:10.1161/01.cir.0000435708.67487.da.
- Lee PH, Macfarlane DJ, Lam T, Stewart SM. Validity of the international physical activity questionnaire short form (IPAQ-SF): A systematic review. Int J Behav Nutr Phys Act 2011;8:115. doi:10.1186/1479-5868-8-115.
- Line 87: The document says: “Furthermore, since during menstrual cycle hormones’ concentration differs maximally between two particular points -early follicular phase, days 5-7, and mid luteal phase, days 21-22 [13-15], we compared participants being in the early follicular phase (n=12) with those in the mid luteal phase (n=11)".
It is really recommended to incorporate the menstrual cycle point in data analysis, since it could probably be a confounding variable. It is true that having taken into account these two phases of the menstrual cycle is a noteworthy point in this research work. However, the number of women included in this comparison is really low (11 women being in the early follicular phase and 12 women being in the mid luteal phase) with respect to the total study population (93 women university students). In a previous study performed by Burrows and cols (2000), women were also split into two groups according to the particular phase of the menstrual cycle (Follicular and Luteal phases). The main difference with this work was that they expand the number of days included in each phase: the follicular phase included from day 5 to 13 of the menstrual cycle, while the luteal phase included from day 15 to 25. This classification may allow having a larger number of women per group, which may increase the significance of statistical analysis.
- Burrows M, Bird S. The physiology of the highly trained female endurance runner. Sports Med Auckl NZ 2000;30:281–300.
- Line 142. The document says: "However, an unexpected result was that the quartile that under-reported BMI the most (Q1) was not the one with the highest BMI suggesting that BMI per se could not identify the less accurate in their answers participants".
In this section, researchers observe that BMI can not be used to identify participants’ answers. A modification in the weight perception affects in a greater degree the BMI value compared to having a wrong height perception. Normally, the participants underestimated their body mass and overestimated their height. Therefore, the group with greater difference in %ΔWeight has been the one with the greater difference in %ΔBMI. The difference in body mass between self-reported and real measure is an indicator more significant than the difference observed in BMI.
Furthermore, the last part of this sentence should be rewritten since it's not well-written and therefore is not easily understandable by the lecturer. Maybe, you would like to say: "could not identify the participants whose answers were less accurate"
- Line 147. The manuscript says: "Inverse small correlation was observed between weekly MET×min and under-reporting weight and BMI indicating that physically active students perceive more accurately their body compared to their less active peers"
It's necessary to perform a statistical comparison of the physical exercise level among groups (Q1 vs Q2 vs Q3 vs Q4) in order to know if there are significant differences and being abole to afirm that the group inlcuding individuals whose answer are more accurate is the one that perform higher amount of physical activities.
Author Response
Comments and Suggestions for Authors
The novelty of this work lies in examining the self-reported perception of body mass and height in women university students (age range between 18 and 30 years), a population with a very specific profile that has not been studied in detail until now. The results presented in this work confirms the hypotheses initially proposed, since women university students tend to over-report height and under-report body mass. Comparison between self-perception and real measures was performed taking into account different conditions, such as the times of the day, the physical activity or the menstrual cycle phase.
This study recommends taking into account the deviations of self-evaluation from real measures by health professionals and researchers when administering questionnaires to women university students. The conclusions and their practical application may be interesting for the readerships. However, the use of questionnaires to evaluate several characteristics related to physical activity and its relationship with other physiological parameters presents some controversy in the scientific field.
The results presented in this study are interpreted appropriately and support the research conclusions. Data analysis and interpretation is adequate to the hypotheses proposed since results obtained are able to answer the different questions proposed at the beginning of the work.
However, the text should be improved by including some details explained below.
Answer: We agree with the expert reviewer and addressed these issues. Please, find our specific answers below.
- Line 80: The document says: “completed the short version of the International Physical Activity Questionnaire (IPAQ) [12] and were tested for anthropometric characteristics. The outcome of IPAQ that was analyzed was the weekly product of metabolic equivalent (MET) and minutes, i.e. MET×min".
In the work carried out by Craig et al (2003), the validation of the IPAQ questionnaire for the quantification of the physical exercise intensity performed by an individual with the METs product and the minutes performed was proposed. In the same way, Strath et al (2013) presents the same methodology to quantify the intensity of exercise carried out based on a questionnaire. However, Lee et al (2011) pointed out the weaknesses of interpreting the results obtained via the use of self-reported questionnaires. Their conclusions stated the low correlation between data collected with questionnaires and the amount of physical activity performed by participants. They showed that self-reported questionnaires overestimate the actual physical activity carried out by participants.
This observation does not directly affect the work done. However, it should be expressed in the main text. It should be clearly explained that the use of questionnaires is a tool for comparison between the different groups defined in each quartile, and it is not a method for quantitatively measure the physical activity carried out by a participant because of the low correlation observed between the self-reported and real measures. For future studies, it would be advisable to use objective methods for measuring physical activity, such as the use of accelerometry.
- Craig CL, Marshall AL, Sjöström M, Bauman AE, Booth ML, Ainsworth BE, et al. International physical activity questionnaire: 12-country reliability and validity. Med Sci Sports Exerc 2003;35:1381–95. doi:10.1249/01.MSS.0000078924.61453.FB.
- Strath SJ, Kaminsky LA, Ainsworth BE, Ekelund U, Freedson PS, Gary RA, et al. Guide to the Assessment of Physical Activity: Clinical and Research Applications A Scientific Statement From the American Heart Association. Circulation 2013:01.cir.0000435708.67487.da. doi:10.1161/01.cir.0000435708.67487.da.
- Lee PH, Macfarlane DJ, Lam T, Stewart SM. Validity of the international physical activity questionnaire short form (IPAQ-SF): A systematic review. Int J Behav Nutr Phys Act 2011;8:115. doi:10.1186/1479-5868-8-115.
Answer: We agree with the expert reviewer and discussed this aspect in the discussion using the recommended references („It should be highlighted that physical activity (weekly MET×min) in the present study was evaluated using the short version of IPAQ, which has been shown previously as a valid and reliable measure [22,23]. Nevertheless, Lee and colleagues [24] identified limitations of interpreting the results obtained via the use of self-reported questionnaires, e.g. low correlation between data collected with questionnaires and the amount of physical activity performed by participants, and overestimation of the actual physical activity. These limitations should be considered when interpreting the results about physical activity. For future studies, it would be advisable to use objective methods for measuring physical activity, e.g. accelerometers.“).
Line 87: The document says: “Furthermore, since during menstrual cycle hormones’ concentration differs maximally between two particular points -early follicular phase, days 5-7, and mid luteal phase, days 21-22 [13-15], we compared participants being in the early follicular phase (n=12) with those in the mid luteal phase (n=11)".
It is really recommended to incorporate the menstrual cycle point in data analysis, since it could probably be a confounding variable. It is true that having taken into account these two phases of the menstrual cycle is a noteworthy point in this research work. However, the number of women included in this comparison is really low (11 women being in the early follicular phase and 12 women being in the mid luteal phase) with respect to the total study population (93 women university students). In a previous study performed by Burrows and cols (2000), women were also split into two groups according to the particular phase of the menstrual cycle (Follicular and Luteal phases). The main difference with this work was that they expand the number of days included in each phase: the follicular phase included from day 5 to 13 of the menstrual cycle, while the luteal phase included from day 15 to 25. This classification may allow having a larger number of women per group, which may increase the significance of statistical analysis.
- Burrows M, Bird S. The physiology of the highly trained female endurance runner. Sports Med Auckl NZ 2000;30:281–300.
Answer: We agree with the expert reviewer and re-analyzed the data using the recommended methodology. Despite the inclusion of more participants using this approach, the findings were „weaker“ and we did not add them in the results. Nevertheless, we discussed this approach in the methods using the recommended reference. („This classification was preferred instead of classifications using wider ranges of menstrual days (e.g. follicular phase, day 5 to 13; luteal phase, day 15 to 25) [16]. Although the later classification would lead in the increase of the sample size, increasing the range of days would decrease the differences between hormones’ concentration and, consequently, their potential effect on weight.“) An explanation might be that „hormones’ concentration during menstrual cycle differs maximally between two particular points - early follicular phase, days 5-7, and mid luteal phase, days 21-22“, whereas using larger ranges of days result in smaller differences in hormones‘ concentration and consequently smaller differences in weight.
Line 142. The document says: "However, an unexpected result was that the quartile that under-reported BMI the most (Q1) was not the one with the highest BMI suggesting that BMI per se could not identify the less accurate in their answers participants".
In this section, researchers observe that BMI cannot be used to identify participants’ answers. A modification in the weight perception affects in a greater degree the BMI value compared to having a wrong height perception. Normally, the participants underestimated their body mass and overestimated their height. Therefore, the group with greater difference in %ΔWeight has been the one with the greater difference in %ΔBMI. The difference in body mass between self-reported and real measure is an indicator more significant than the difference observed in BMI.
Furthermore, the last part of this sentence should be rewritten since it's not well-written and therefore is not easily understandable by the lecturer. Maybe, you would like to say: "could not identify the participants whose answers were less accurate"
Answer: We agree with the expert reviewer and revised this part („A potential explanation of the underestimation of weight might be a recent gain of weight, since entering university has been identified as a major cause of gain of weight in women [20]. Surprisingly, the group who under-reported BMI the mostly (Q1) did not have the largest BMI, which was in disagreement with previous studies, in which heavier women students tended to under-report their weight [21,22]. This discrepancy might be explained by the absence of high scores of BMI in the participants of the present study; on the other hand, the underestimation of BMI was observed previously in individuals at the high end of BMI, particularly at values larger than 28 kg.m-2 [22].“).
Line 147. The manuscript says: "Inverse small correlation was observed between weekly MET×min and under-reporting weight and BMI indicating that physically active students perceive more accurately their body compared to their less active peers"
It's necessary to perform a statistical comparison of the physical exercise level among groups (Q1 vs Q2 vs Q3 vs Q4) in order to know if there are significant differences and being able to affirm that the group including individuals whose answer are more accurate is the one that perform higher amount of physical activities.
Answer: We agree with the expert reviewer and we already presented this aspect in the results („A medium effect of %ΔBMI on MET×min was observed (p=0.086, η2=0.127) with Q3 the most physically active and Q4 the least“).
Reviewer 3 Report
BRIEF SUMMARY
This article focuses mainly on examining the validity of self-reported body mass, height and BMI in women students The study of this topic can be useful and interesting when making studies where subjects self-report their anthropometric data. However, this circumstance has been widely developed in the previous specific literature, obtaining similar results to this study even in the same sex and age segments. In addition, in the opinion of this reviewer, the characteristics of the sample included in the study may bias their results. The main novelty of this research is to introduce complementary variables such as physical activity, time of day and menstrual cycle phase. Equally, the adequacy of the methodological procedures used must be recognized.
More details are provided below.
INTRODUCTION.
Much of the introduction is appropriate. The antecedents on the subject to be investigated are clearly stated and an attempt is made to justify the need to study it in university women. The objectives and the corresponding hypotheses are correctly formulated. However, in relation to the secondary objective of to analyze the role of physical activity, time of day and menstrual cycle phase on this validity, a theoretical justification of this need is missed.
MATERIALS AND METHODS
Some considerations about the section are the following:
· The career that the participants are studying can significantly bias the results. The training received for the topic as well as their concerns about health are crucial when assessing and interpreting their anthropometric data. This reviewer estimates that the results are difficult to generalize.
· Participants were grouped into quartiles according to percentage difference in reporting BMI but not categorized according to their actual weight status: low weight, normal weight, overweight and obesity.
· Another unspecified aspect that significantly affects the results is satisfaction or dissatisfaction with one's body.The body image can greatly condition the perceptions of the measurements of our body, to the point of creating corporal perceptual disorders.
· Parametric tests have been carried out for the analysis of the data, when homoscedasticity or normality tests have not been carried out. When the sample of 93 subjects is analyzed from their quartiles (n = 23) it is possible that the subgroups do not adapt to a normal curve.
RESULTS
The results are well presented, although their interpretation should be considered with caution.
· The comments in the previous section (methodology) can affect the results decisively.
· If Participants were grouped into quartiles according to percentage difference in reporting BMI, one doubt arises from the reading of table 1. Why do participants in quartile 2 (24.2±4.4) have higher BMI than those in Q1 (23.3±3.9), while the data in Q3 and Q4 are lower?
DISCUSSION
· The main aim of the study is to examine the validity of self-reported body mass, height and BMI in women students. The results again show that women student over-reported height, and under-reported weight and BMI. In this sense, the study brings little novelty to the study area.
· On the other hand, the overestimation of height and underestimation of weight was in agreement with previous findings in university students, however, it does not compare to what extent this overestimation or underestimation is greater or less than in other studies.
· The limitations and conclusions are well argued. I fully agree with the limitations of the study presented by the authors.
Author Response
Comments and Suggestions for Authors
BRIEF SUMMARY
This article focuses mainly on examining the validity of self-reported body mass, height and BMI in women students. The study of this topic can be useful and interesting when making studies where subjects self-report their anthropometric data. However, this circumstance has been widely developed in the previous specific literature, obtaining similar results to this study even in the same sex and age segments. In addition, in the opinion of this reviewer, the characteristics of the sample included in the study may bias their results. The main novelty of this research is to introduce complementary variables such as physical activity, time of day and menstrual cycle phase. Equally, the adequacy of the methodological procedures used must be recognized.
Answer: We agree with the expert reviewer and highlighted the novel aspects of this study in the discussion (physical activity, time of day and menstrual cycle phase).
More details are provided below.
INTRODUCTION
Much of the introduction is appropriate. The antecedents on the subject to be investigated are clearly stated and an attempt is made to justify the need to study it in university women. The objectives and the corresponding hypotheses are correctly formulated. However, in relation to the secondary objective of to analyze the role of physical activity, time of day and menstrual cycle phase on this validity, a theoretical justification of this need is missed.
Answer: We agree with the expert reviewer and added the rationale or the secondary objectives in the end of the introduction („Furthermore, information about the role physical activity, time-of-the-day and menstrual cycle on the accuracy of self-reporting these anthropometric characteristics would be of practical value when administering and interpreting a survey.“).
MATERIALS AND METHODS
Some considerations about the section are the following:
The career that the participants are studying can significantly bias the results. The training received for the topic as well as their concerns about health are crucial when assessing and interpreting their anthropometric data. This reviewer estimates that the results are difficult to generalize.
Answer: We agree with the expert reviewer and mentioned this aspect in the limitations of the study („A limitation of the present study was that the sample consisted only of students of Biomedical Sciences. It was assumed that students in health-allied professions would be more aware about their anthropometric characteristics than students of other professions.“).
Participants were grouped into quartiles according to percentage difference in reporting BMI but not categorized according to their actual weight status: low weight, normal weight, overweight and obesity.
Answer: We agree with the expert reviewer and added this observation and a justification in the methods („Four participants (4.3% of the sample) were under-weight, 67 were normal-weight (72.0%), 18 were overweight (19.4%) and four were obese (4.3%).“ „A comparison of BMI groups (under-, normal-, over-weight and obese) was not performed due to large difference in sample size of each group.“).
Another unspecified aspect that significantly affects the results is satisfaction or dissatisfaction with one's body. The body image can greatly condition the perceptions of the measurements of our body, to the point of creating corporal perceptual disorders.
Answer: We agree with the expert reviewer and discussed this topic in the discussion („Moreover, an unspecified aspect that was not addressed in this study and might affect the results was satisfaction or dissatisfaction with one's body, since body image could condition the perceptions of the measurements of human body [31].“).
Parametric tests have been carried out for the analysis of the data, when homoscedasticity or normality tests have not been carried out. When the sample of 93 subjects is analyzed from their quartiles (n = 23) it is possible that the subgroups do not adapt to a normal curve.
Answer: We agree with the expert reviewer and added this information in the methods section („Data - for the whole sample and not for each sub-group - were tested for normality using the Kolmogorov-Smirnov test and visual inspection of quantile-quantile plots.“).
RESULTS
The results are well presented, although their interpretation should be considered with caution.
The comments in the previous section (methodology) can affect the results decisively.
Answer: We agree with the expert reviewer and commented in the interpretation of the findings in the discussion („Thus, caution would be needed to generalize our findings to students of other professions“).
If Participants were grouped into quartiles according to percentage difference in reporting BMI, one doubt arises from the reading of table 1. Why do participants in quartile 2 (24.2±4.4) have higher BMI than those in Q1 (23.3±3.9), while the data in Q3 and Q4 are lower?
Answer: We agree with the expert reviewer and discussed this finding in the discussion („Surprisingly, the group who under-reported BMI the mostly (Q1) did not have the largest BMI, which was in disagreement with previous studies, in which heavier women students tended to under-report their weight [11,21,22]. This discrepancy might be explained by the absence of many high scores of BMI in the participants of the present study; on the other hand, the largest underestimation of BMI was observed previously in individuals at the high end of BMI, particularly at values larger than 28 kg.m-2 [22].“). We double-checked our data, and re-analyzed them; all data were presented correctly.
DISCUSSION
The main aim of the study is to examine the validity of self-reported body mass, height and BMI in women students. The results again show that women student over-reported height, and under-reported weight and BMI. In this sense, the study brings little novelty to the study area.
Answer: We agree with the expert reviewer; however, the novelty is in the factors that may influence this trend, i.e. time-of-the-day, physical activity and menstrual cycle.
On the other hand, the overestimation of height and underestimation of weight was in agreement with previous findings in university students, however, it does not compare to what extent this overestimation or underestimation is greater or less than in other studies.
Answer: We agree with the expert reviewer and quantified these comparisons („The overestimation of height (by ~1%), and underestimation of weight (by ~1%) and BMI (by ~3%)...“ „For instance, American women students over-reported height by 0.5 cm, and under-reported weight and BMI by 1.3 kg and 0.6, respectively [11].“).
The limitations and conclusions are well argued. I fully agree with the limitations of the study presented by the authors.
Answer: We agree with the expert reviewer.
Reviewer 4 Report
Overall, the study’s objectives and results were interesting. The authors made several minor grammatical errors throughout the paper that can be improved through detailed proof reading. The following are a few suggestions for the authors to consider:
Introduction, line 40: There appear to be a word missing after the word “older” in this sentence: “. . . for instance, women and older reported more accurately their . . .”
Introduction, line 45: Insert the word “of” after the word “consist” in this sentence: “University students consist a population group with specific characteristics . . .”
Introduction, line 50: Insert the word “on” after the word “depend” in this sentence: “. . . characteristics might depend time of day”
Methods, line 91: Can you explain this sentence further: “Also, all participants were grouped into quartiles according to percentage difference in reporting BMI (%ΔBMI).” Specifically, explain how the participants were split into the 4 quartiles in table 1.
Methods, line 96: Would you explain why “Statistical significance was set at p=0.10”? Typically, p is set at 0.05.
Results, line 116: Change “accurately” to “accurate” in this sentence: “Participants being in the early follicular phase reported likely more accurately. . .”
Results, line 129: Remove the word “in” before the word “classified” in this sentence: “ . . where at 12-2pm most participants were in classified in Q3. . .”
Discussion, line 140: Change “accurately” to “accurate” in this sentence: “. . . reported likely more accurately weight and BMI than . . .”
Table 1: It is unclear what is meant by quartile groups. Please explain in text and/or in the table.
Author Response
Comments and Suggestions for Authors
Overall, the study’s objectives and results were interesting. The authors made several minor grammatical errors throughout the paper that can be improved through detailed proof reading. The following are a few suggestions for the authors to consider:
Introduction, line 40: There appear to be a word missing after the word “older” in this sentence: “. . . for instance, women and older reported more accurately their . . .”
Answer: We agree with the expert reviewer and corrected it adding “respondents”.
Introduction, line 45: Insert the word “of” after the word “consist” in this sentence: “University students consist a population group with specific characteristics . . .”
Answer: We agree with the expert reviewer; however, we changed “consist” to “comprise” according to the comment of reviewer 1.
Introduction, line 50: Insert the word “on” after the word “depend” in this sentence: “. . . characteristics might depend time of day”
Answer: We agree with the expert reviewer and corrected it as suggested.
Methods, line 91: Can you explain this sentence further: “Also, all participants were grouped into quartiles according to percentage difference in reporting BMI (%ΔBMI).” Specifically, explain how the participants were split into the 4 quartiles in table 1.
Answer: We agree with the expert reviewer and added this information in the methods section (“with Q1 underreporting BMI the most. %ΔBMI was calculated using the formula 100×(reported BMI-measured BMI)/measured BMI with negative or positive scores denoting under-reporting or over-reporting BMI, respectively.”).
Methods, line 96: Would you explain why “Statistical significance was set at p=0.10”? Typically, p is set at 0.05.
Answer: We agree with the expert reviewer and explained the choice („due to the relatively small size of sub-groups“).
Results, line 116: Change “accurately” to “accurate” in this sentence: “Participants being in the early follicular phase reported likely more accurately. . .”
Answer: We agree with the expert reviewer and corrected it as suggested.
Results, line 129: Remove the word “in” before the word “classified” in this sentence: “ . . where at 12-2pm most participants were in classified in Q3. . .”
Answer: We agree with the expert reviewer and corrected it as suggested.
Discussion, line 140: Change “accurately” to “accurate” in this sentence: “. . . reported likely more accurately weight and BMI than . . .”
Answer: We agree with the expert reviewer and corrected as suggested.
Table 1: It is unclear what is meant by quartile groups. Please explain in text and/or in the table.
Answer: We agree with the expert reviewer and clarified it in the methods section and under table 1.
Round 2
Reviewer 3 Report
Although this reviewer maintains the opinion that some of the problems presented by the article are difficult to improve as the characteristics of the sample and the generalization of the results, the manuscript has improved with respect to the initial version. The improvements carried out from the initial comments, both this and the other reviewers are good, so I think the research could be published.
Reviewer 4 Report
The authors made the suggested changes to the paper.